# Seawater Culture Increases Omega-3 Long-Chain Polyunsaturated Fatty Acids (N-3 LC-PUFA) Levels in Japanese Sea Bass (*Lateolabrax japonicus*), Probably by Upregulating Elovl5

**DOI:** 10.3390/ani10091681

**Published:** 2020-09-17

**Authors:** Xiaojing Dong, Jianqiao Wang, Peng Ji, Longsheng Sun, Shuyan Miao, Yanju Lei, Xuedi Du

**Affiliations:** 1Laboratory of Aquaculture Nutrition and Feed, College of Animal Science and Technology, Yangzhou University, Yangzhou 225009, China; xjdong@yzu.edu.cn (X.D.); jian_qiao_wang@163.com (J.W.); jp2278362123@163.com (P.J.); lssun@yzu.edu.cn (L.S.); shuyanmiao@126.com (S.M.); 2Collaborative Innovation Center for Efficient and Health Production of Fisheries in Hunan Province, Key Laboratory of Health Aquaculture and Product Processing in Dongting Lake Area of Hunan Province, Zoology Key Laboratory of Hunan Higher Education, Hunan University of Arts and Science, Changde 415000, China; leiyanju2009@163.com

**Keywords:** sea bass, freshwater pond culture, marine cage culture, n-3 LC-PUFA, gene promoter methylation

## Abstract

**Simple Summary:**

Japanese sea bass can be farmed in both marine and inland waters, and marine cage cultured fish have a higher polyunsaturated fatty acid content and tastes better. However, how marine cage rearing improves its polyunsaturated fatty acid content has not been clarified yet. Thus, we investigated the role of fatty acid desaturase 2 and fatty acid elongase 5, two key players in fatty acid metabolism, in long-chain polyunsaturated fatty acids biosynthesis in Japanese sea bass under seawater culture. We found that the content of eicosapentaenoic acids (EPA) and docosahexaenoic acids (DHA) in the seawater group was significantly higher than in the freshwater group, and the *fatty acid elongase 5* gene expression level in the seawater group was significantly higher than in the freshwater group. On the other hand, however, the *fatty acid desaturase 2* expression level in the seawater group was significantly lower than in the freshwater group. A further comparison of gene promoter methylation patterns of *fatty acid desaturase 2* and *fatty acid elongase 5* showed that seawater reared fish were less methylated in the *fatty acid elongase 5* promoter region, but far more methylated in *fatty acid desaturase 2* promoter region. Taken together, our study suggests important roles of fatty acid elongase 5 in the enhanced biosynthesis of long-chain polyunsaturated fatty acids under marine cage culture compared with freshwater pond culture.

**Abstract:**

The fatty acid compositions of the fish muscle and liver are substantially affected by rearing environment. However, the mechanisms underlying this effect have not been thoroughly described. In this study, we investigated the effects of different culture patterns, i.e., marine cage culture and freshwater pond culture, on long-chain polyunsaturated fatty acids (LC-PUFA) biosynthesis in an aquaculturally important fish, the Japanese sea bass (*Lateolabrax japonicus*). Fish were obtained from two commercial farms in the Guangdong province, one of which raises Japanese sea bass in freshwater, while the other cultures sea bass in marine cages. Fish were fed the same commercial diet. We found that omega-3 long-chain polyunsaturated fatty acids (n-3 LC-PUFA) levels in the livers and muscles of the marine cage cultured fish were significantly higher than those in the livers and muscles of the freshwater pond cultured fish. Quantitative real-time PCRs indicated that fatty acid desaturase 2 (*FADS2*) transcript abundance was significantly lower in the livers of the marine cage reared fish as compared to the freshwater pond reared fish, but that fatty acid elongase 5 (*Elovl5*) transcript abundance was significantly higher. Consistent with this, two of the 28 CpG loci in the *FADS2* promoter region were heavily methylated in the marine cage cultured fish, but were only slightly methylated in freshwater pond cultured fish (n = 5 per group). Although the *Elovl5* promoter was less methylated in the marine cage reared fish as compared to the freshwater pond reared fish, this difference was not significant. Thus, our results might indicate that *Elovl5*, not *FADS2*, plays an important role in the enhancing LC-PUFA synthesis in marine cage cultures.

## 1. Introduction

It has frequently been shown that the lipid and fatty acid composition of aquatic animals is influenced by the culture environment [1,2,3,4,5,6,7,8,9], and the change in the biosynthesis of long-chain polyunsaturated fatty acids (LC-PUFA) has been attributed to water salinity regulation [9,10]. Studies of this process have recovered inconsistent results: higher salinities have been associated with both increased [4,5,9] and decreased [1,2] levels of LC-PUFA in different fish species. These discrepancies may be because different fish species have adapted to their local environments by different means. Irrespective of these discrepancies, it is clear that environmental factors affect LC-PUFA levels in fish. It is important to investigate the mechanisms by which the culture environment influences fish LC-PUFA levels for two reasons. First, LC-PUFA content is used to assess the nutritional value of fish [3,4,11]. Second, fish are the primary source of LC-PUFA for humans [12], and LC-PUFA plays numerous physiologically important roles essential to human health in a variety aspects, such as inflammation, depression, coronary heart disease, and so on [13,14].

As a euryhaline fish species, Japanese sea bass *Lateolabrax Japonicus* can be farmed in both marine and inland waters [15]. It has long been one of most common aquaculture fish species in China; China produced over 150,000 and 450,000 tons of Japanese sea bass by marine cage culture and freshwater pond culture, respectively, in 2017 [16]. Previous research has shown that seawater reared sea bass, compared to freshwater pond reared sea bass, showed significantly higher eicosapentaenoic acids (EPA, 20:5n-3) and docosahexaenoic acids (DHA, 22:6n-3) contents in both liver and muscle tissues [4]. To date, however, the underlying mechanism has not been well studied. Since the liver serves as the main organ responsible for fatty acids biosynthesis in fish, it is conjectured that biosynthesis of LC-PUFA is probably enhanced in seawater cultured sea bass compared with freshwater reared ones.

The biosynthesis of LC-PUFA in vertebrates, including fish, involves the alternating desaturation and elongation of the C18 fatty acid. Fatty acid Δ6 desaturase (FADS2) and fatty acid elongase 5 (Elovl5) catalyze the first desaturation step and the first elongation step, respectively; these enzymes, thus, play vital roles in LC-PUFA biosynthesis [17,18,19,20]. It has been shown that the ambient environment and nutritional factors regulate the desaturation and elongation pathways of LC-PUFA synthesis in fish [9,10,21]. The nutritional modulation of FADS2 and Elovl5 have been extensively researched to test whether desaturase and elongase activity could compensate for the reduction of tissue LC-PUFA levels due to supplementation of dietary vegetable oil [22,23]. Some studies have focused on the regulatory mechanisms of *FADS2* and *Elovl5* genes. For instance, some transcription factors, Sterol Regulatory Element Binding Protein-1 (SREBP-1) and Peroxisome Proliferator-Activated Receptors (PPARs), regulate the two genes [24,25]. In addition to nutritional factors, environmental factors have been demonstrated to mediate *FADS2* gene expression regulation in fish. In Atlantic salmon, for instance, an increase in LC-PUFA biosynthesis was observed during parr–smolt transformation via increasing *FADS2* gene expression [26]. To date, however, it has been unclear whether Elovl5 is associated with environmental factors regulated LC-PUFA biosynthesis in fish, and, if so, how.

At present, many researchers are focused on uncovering the mechanism underlying phenotypic variation in response to different environments. Growing evidence indicates that DNA methylation plays an important role in helping animals cope with different environments [27], which predominantly occurs at the 5-position of cytosine, and plays a vital role in the epigenetic control of gene expression [28]. It was found that exposing adult zebrafish to estrogens could led to a decrease in *Vitellogenin* gene promotor methylation level, and hence, increase gene expression [29]. In Half smooth tongue sole, water salinity stress may regulate gene expression via changing DNA methylation patterns at tissue-specific epigenetic loci [30]. Therefore, in this study, we aimed to answer the question that why seawater cultured sea bass owns higher EPA and DHA contents compared with freshwater cultured fish by focusing on key genes involved in LC-PUFA biosynthesis from the aspect of gene expression and epigenetic regulation.

## 2. Materials and Methods

### 2.1. Ethic Statement

Our study on Japanese sea bass was approved by the Animal Care and Use Committee of the Yangzhou University (ethical protocol code: YZUDWSY 2017-09-06), and all efforts were made to minimize suffering. Experimental fish was first anesthetized with MS-222 (200 mg/L, Sigma, St. Louis, MO, USA) before sacrificing and handling.

### 2.2. Experimental Fish and Diets

Adult Japanese sea bass were cultured in marine cages (salinity 26.2 g/L, MC group, Berlin, Germany) and freshwater pond (salinity 0.1 g/L, FP group) at farms in Guangdong, China (in Shantou and Zhuhai, respectively). The two farms used the same fish fry from a common breeding company, and fed identical diets (8985; TongWei, Chengdu, China; Table 1) throughout the entire culture period. The FP sea bass was cultured in pond of 1.5 acres in size, while the MC fish was raised in marine cages. Although the marine cages were much smaller in size (10 m × 10 m × 10 m) compared with the freshwater pond, breeding density in the two culture patterns was comparable, approximately 6–8 per cubic meter at harvest. Experimental fish had been reared for eight months before sampling in December 2018, and weighed 400–500 g.

### 2.3. Sampling

After fasting and anesthetization, five fish from each group were dissected, and the back muscle and liver were collected from each fish. Tissues were stored at −20 °C for DNA extraction and fatty acid composition analysis. For RNA isolation and gene expression analysis, the livers were removed from another five fish per group, immediately frozen in liquid nitrogen, and stored at −80 °C until use.

### 2.4. Proximate Muscle Composition

The dry matter, crude protein and crude lipid contents of all muscle samples were determined following previously described methods [31,32].

### 2.5. Fatty Acid Composition

All samples subjected to fatty acids analysis were first freeze-dried at −40 °C for 48 h, and then smashed and homogenized with a high-throughput and high-speed grinding machine (Xinzhi biotechnology Co. Ltd., Ningbo, China). For each sample, a moderate amount of powder was removed to hydrolysis in 10 mL hydrochloric acid solution with 100 mg gallic acid and 2 mL ethanol at 70–80 °C for 40 min. The hydrolysate was further extracted three times using ether and petroleum ether mixture and steamed to dry at 100 ± 5 °C for 2 h followed by saponification and esterification in alkaline solutions at 85 °C in water bath. After cooling down to room temperature, 1 mL n-hexane was added to extract the esterification product, and 100 μL supernatant was drawn and stabilized to 1 mL with n-hexane before filtration with 0.45 μm membrane. The fatty acid compositions were then determined with a gas chromatograph (7890A; Agilent, Agilent Technologies (Beijing) Co. Ltd., Beijing, China), using a CNW CD-2560 chromatographic column (100 m × 0.25 mm × 0.20 μm, Chemicals Northwest, Warrington, UK). The heating procedure was as follows: 130 °C for 15 min, then raised to 240 °C at a rate of 4 °C/min for 30 min. The injector and detector temperatures were set to 250 °C and 260 °C, respectively, and the flow rate of carrier gas was 0.5 mL/min. The split ratio was 10:1. Fatty acid quantification was normalized against standard mixture of 35 fatty acid methyl esters (Sigma, CRM47885, Merck Chemical Technology (Shanghai) Co. Ltd., Shanghai, China) and calculated using the following formula:
(1)$$W=\frac{C\;\times V\;\times N}{m}\times k$$ where W is every fatty acid content in sample (mg/kg), C is the concentration of methyl esters of fatty acids (mg/L), V is the constant volume (mL), N is the dilution multiple, k is the conversion coefficient of each fatty acid methyl ester to fatty acid and m is the weight of the sample (g).

### 2.6. Quantitative Real-Time PCR Analysis

Total RNA was extracted from each liver sample using Trizol reagent (Invitrogen, CA, USA). Total RNA quality was tested with electrophoresis, using a 1.2% denaturing agarose gel. cDNA synthesis and quantitative real-time PCR were performed as previously described [33]. Briefly, the concentration of the prepared total RNA was determined and 1 μg of total RNA was treated with gDNA Eraser (Takara, Takara Biotechnology (Dalian) Co. Ltd., Dalian, China) to remove possible DNA contaminants according to the manufacturer’s instructions. Purified RNA was subjected to reverse transcription using the PrimeScript RT reagent kit (Takara) following the manufacturer’s instructions. Quantitative real-time PCR was performed on AB 7500 Fast platform (Applied Biosystems, Carlsbad, CA, USA) according to the instructions provided in the fluorescence quantitative PCR kit (Takara). Primers for β-actin were from our former research, while primers for FADS2, and Elovl5 were designed using Oligo software v.7 based on mRNA sequences downloaded from GenBank database (Table 2). Primer amplification efficiencies for *β-actin*, *FADS2* and *Elovl5* were 95.7%, 101.2% and 98.3%, respectively, determined by quantitative real time PCR analysis on gradient diluted cDNA. Cycle number values were normalized against the reference gene (β-actin). Expression data were analyzed using the 2^−ΔΔCt^ relative quantification method and the expression value represented the n-fold difference relative to the calibration [34].

### 2.7. Cloning of the Elovl5 Promoters

We performed restriction endonuclease reactions using the genomic DNA extracted from each liver sample. Each 20 μL reaction volume contained 400 ng DNA, 2 μL EcoR I and 2.5 μL 10 × reaction buffer. The reaction mixture was thermally insulated at 37 °C for 5 h, and then inactivated at 75 °C for 10 min. To perform ligation, we combined the reaction mixture with 1 µL T4 ligase, 2.5 μL T4 buffer and 1 μL adapter (Sangon Biotech, Shanghai, China) in a 45 μL volume. The ligation mixture was incubated at 4 °C overnight before use.

Based on the *Elovl5* gene cloned from Japanese sea bass (KY688066.1), two reverse specific primers were designed (SP1 and SP2; Table 2). We then cloned the *Elovl5* gene promoter with two-round PCR, using SP1 and SP2 with the forward adapter primers AP1 and AP2 (Table 2), respectively. The cycling conditions for the first round PCR were as follows: preheating at 95 °C for 5 min; 10 cycles of amplification (94 °C for 30 s, 68 °C for 30 s, ramping at −0.8 °C/cycle to 72 °C and 72 °C for 150 s); 30 cycles of amplification (94 °C for 30 s, 60 °C for 30 s and 72 °C for 150 s); and elongation at 72 °C for 10 min. The cycling conditions for the second-round PCR were as follows: 28 cycles of amplification (94 °C for 30 s, 58 °C for 30 s and 72 °C for 150 s), followed by elongation at 72 °C for 10 min. The purified PCR products were cloned into pMD18-T vectors (TaKaRa Biotechnology, Dalian, China) and Sanger-sequenced.

### 2.8. Bisulfate Genomic Sequencing (BSP)

DNA was extracted from the liver samples of the experimental fish. Sodium bisulfate modifications were performed with the EZ DNA Methylation-Gold Kit D5005 (Zymo Research, Irvine, CA, USA), following the manufacturer’s instructions. After bisulfite conversion, the core regions of the FADS2 and Elovl5 gene promoters (Figure 1) were PCR amplified using primers specific to the FADS2 or Elovl5 gene sequence (Table 2). The FADS2 promoter sequence has been previously published [18]. PCRs were performed in 50 μL volumes, each containing 2 µL bisulfate-modified DNA, 0.5 μL Taq polymerase, 2 μL primers, 2 μL dNTP and 5 μL Taq Buffer (with MgCl2). The mixture was preheated at 95 °C for 5 min, then subjected to 35 cycles of amplification (95 °C for 30 s, 58 °C for 30 s and 72 °C for 30 s). The purified PCR products were cloned into pMD18-T vectors (TaKaRa Biotechnology), and 10 clones from each sample were randomly selected for Sanger sequencing.

### 2.9. Statistical Analyses

Statistical analyses were performed using SPSS 16.0 (IBM Corporation, Amonk, NY, USA). All results are presented as means ± standard deviation. All data were analyzed using one-way analyses of variance (ANOVAs). We determined whether differences between means were significant using Tukey’s multiple range tests. We considered *p* < 0.05 statistically significant. MethPrimer (http://www.urogene.org/methprimer) [35] was used to identify the CpG islands.

## 3. Results

### 3.1. Proximate Composition of the Muscle

The crude protein and crude lipid contents of the muscle were significantly higher in the MC group as compared to the FP group (*p* < 0.05, Table 3); however, moisture content was significantly lower in the MC group (*p* < 0.05, Table 3).

### 3.2. Fatty Acid Composition of the Muscle and of the Liver

Our analysis of fatty acids in the muscle showed that all fatty acid except C18:3n-6 were significantly more abundant in the MC group fish muscle as compared to the FP group fish muscle (*p* < 0.05, Table 4); levels of n-3 LC-PUFA were particularly high in the MC group fish muscle. In addition, both C20:5n-3 (EPA) and C22:6n-3 (DHA) were more than three times more abundant in the MC group fish muscle than in the FP group fish muscle. In contrast, most of the fatty acids in the MC group livers were significantly less abundant that those in the FP group livers (*p* < 0.05, Table 5). The only exceptions were EPA and DHA, which were much more abundant in the MC group than in the FP group. No significant differences between the two groups were detected for C16:1, C18:1n-9 and C20:4n-6 in the liver (*p* > 0.05).

### 3.3. Gene Expression in the Liver

We identified a significant difference in the abundances of *FADS2* and *Elovl5* transcripts between the two groups (Figure 2). *Elovl5* transcripts in the MC group were significantly more abundant than in the FP group (*p* < 0.05). In contrast, *FADS2* gene expression was significantly lower in the MC group as compared the FP group (*p* < 0.05).

### 3.4. CpG Methylation of the FADS2 and Elovl5 Promoters

The *FADS2* promoter sequence has been previously published [18]. Two CpG islands were identified around the transcriptional start site, but only one had a significantly different rate of methylation between the two groups. Two CpG loci in this region were heavily methylated in the MC group, but were only slightly methylated in the FP group (Figure 3). The CpG methylation rate was significantly negatively correlated with the mRNA expression of *FADS2* (r = −1).

We cloned a 1348-bp fragment upstream of the Elovl5 translation start codon. Only one CpG island was identified between −372-bp and −229-bp upstream of the transcription initiation site. There were no significant differences in the methylation states of the 16 CpG sites in the *Elovl5* gene promoter between the FP group and the MC group (Figure 4).

## 4. Discussion

In some euryhaline fish species, seawater rearing tends to increase the crude protein and crude lipid contents of the muscle, but tends to decrease the muscle moisture content [4,36,37,38]. Our comparisons of the biochemical compositions of marine cage cultured and freshwater pond reared sea bass were consistent with these previous studies. Differences in fish muscle composition between the two culture patterns might be attributed to the lower amount of energy required for osmoregulation in seawater as compared to freshwater [4].

Recent studies have investigated the relationship between LC-PUFA biosynthesis and environmental factors. In Atlantic salmon, LC-PUFA biosynthesis and FADS2 gene expression increased during parr-smolt transformation [26]. However, the reverse was observed in *Siganus canaliculatus*: fish raised in higher salinity waters had lower levels of n-3 LC-PUFA, as well as lower FADS2 mRNA expression in the liver [39]. In this study, noticeable differences in the fatty acid profiles of the muscle and liver tissues were detected between the MC group and the FP group. In the muscle, most fatty acids were significantly more abundant in the MC group than in the FP group. In the liver, however, most of the fatty acids tested were significantly less abundant in the MC group as compared to the FP group. Nonetheless, the levels of EPA and DHA, two of the most important n-3 LC-PUFAs, were higher in both the muscles and the livers of the MC group. These results were similar to those of previous studies of sea bass [1,4].

It has been shown that environmental factors regulate the desaturation pathways involved in LC-PUFA biosynthesis in some fish species [21], but it is unclear whether fatty acid elongation could also be regulated by environmental factors. In *S. canaliculatus*, for example, FADS2 gene expression in the livers of fish inhabiting waters with lower levels of salinity (10 ppt) was 1.56-fold greater compared with that of fish inhabiting waters with normal salinity (32 ppt) [39]. Similarly, in Atlantic salmon, LC-PUFA synthesis is regulated by environmental factors through the modulation of the expression levels of genes encoding the fatty acyl desaturase, but not the expression levels of genes encoding fatty acyl elongase [26]. Here, FADS2 was significantly downregulated in the MC group as compared to the FP group. In contrast, Elovl5 transcripts were significantly more abundant in the MC group, which was consistent with the contents of n-3 LC-PUFA in both liver and muscle. Taking into account the significantly lower contents of 14-18C fatty acids in the liver in MC group than in FP group, we might guess that the improved EPA and DHA contents in MC group is related to the enhanced fatty acid elongation activity. In other words, that Elovl5, but not FADS2, plays a major role in LC-PUFA biosynthesis in sea bass under seawater conditions.

Salinity stress often alters DNA methylation levels in plants and animals [28,30,40,41]; changes in methylation may help organisms to develop, grow, and survive [42]. To investigate how the expression levels of *FADS2* and *Elovl5* are regulated in marine cage cultured sea bass, the DNA methylation states of the CpG dinucleotides in the gene promoter regions of both genes were determined using BSP. We found that two CpG loci in the *FADS2* gene promoter region were heavily methylated in the MC group, but only slightly methylated in the FP group. More importantly, a significantly negative correlation was identified between the methylation rate and the level of *FADS2* mRNA expression in the liver, suggesting that the down-regulated *FADS2* expression in MC group was probably mediated by gene promoter methylation. In contrast, the methylation rate of CpG loci in the *Elovl5* promoter region was lower in the MC group, although no significant difference was detected. Here, we suggest the existence of a gene network that acts on *Elovl5*, which is likely in response to mainly water salinity, and enhances *Elovl5* gene expression. To test this hypothesis, a well-controlled feeding trial is in progress, in which sea bass is cultured in tanks with different water salinity, and a comparative transcriptome sequencing will be conducted to determine the genes involved in enhancing n-3 LC-PUFA synthesis, especially through regulating *Elovl5* expression, under high water salinity.

## 5. Conclusions

Our findings suggest that marine cage culture of Japanese sea bass can significantly increase the content of n-3 LC-PUFA compared with freshwater pond culture. Taking into account the expression levels of *FADS2* and *Elovl5*, two key players in fatty acid synthesis, and the methylation patterns of the gene promoter regions, we infer that Elovl5, not FADS2, likely plays a major role in enhancing sea bass n-3 LC-PUFA biosynthesis under marine cage culture conditions. Further studies are needed to elucidate the exact mechanism involved.

## Figures and Tables

**Figure 1 animals-10-01681-f001:**
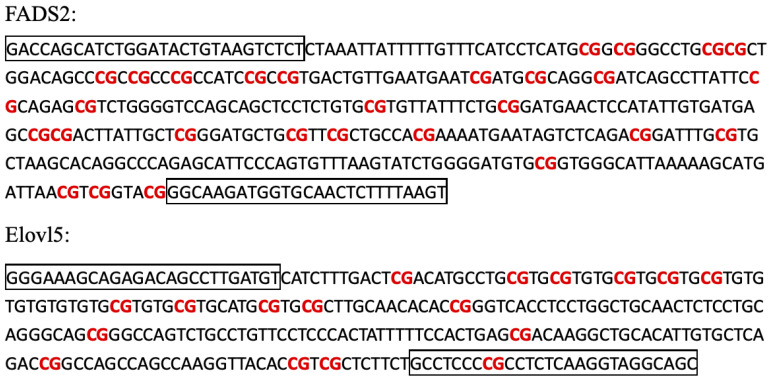
The *FADS2* and *Elovl5* promoter fragments used in BSP. The primers are in the boxes. There are 28 and 17 CpG loci in sequencing area from FADS2 and Elovl5 promoters, respectively, which are marked in red.

**Figure 2 animals-10-01681-f002:**
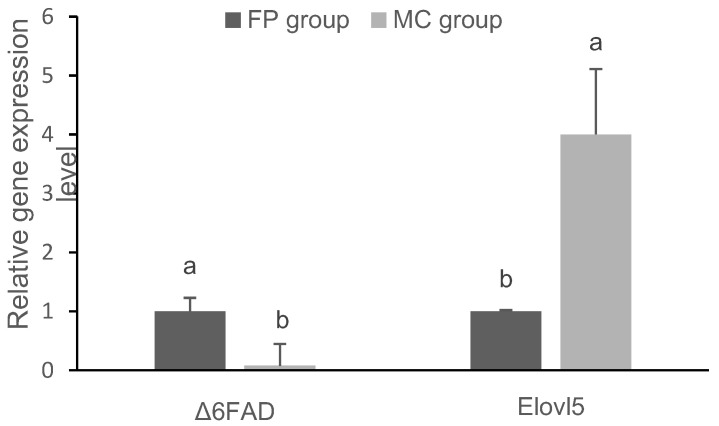
Effects of marine cage culture on *FADS2* and *Elovl5* expression in the liver of Japanese sea bass. Bars with different letter are significantly different (a,b *p* < 0.05; Tukey’s test) between the two groups.

**Figure 3 animals-10-01681-f003:**
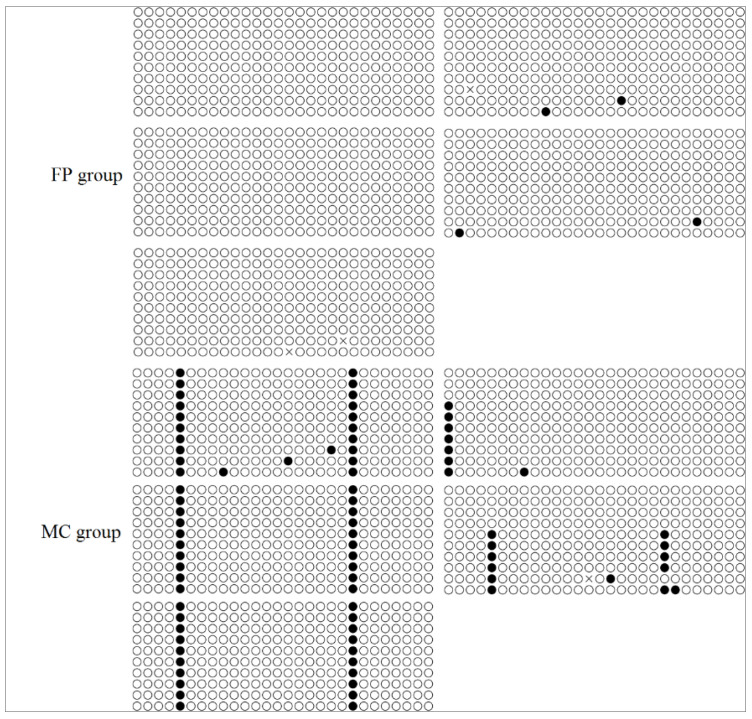
The methylation level of CpG loci in *FADS2* promoter under freshwater and seawater. The methylated and unmethylated CpG loci are shown in black and open circles, respectively.

**Figure 4 animals-10-01681-f004:**
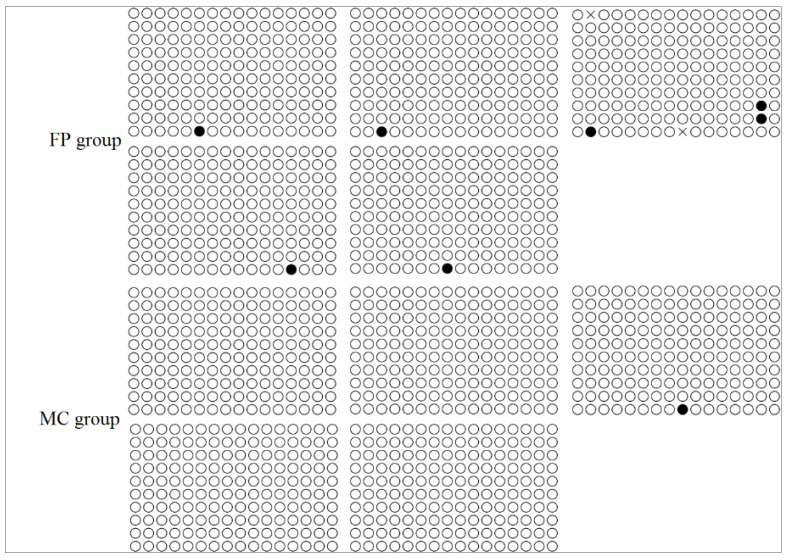
The methylation level of CpG loci in *Elovl5* promoter under freshwater and seawater. The methylated and unmethylated CpG loci are shown in black and open circles, respectively.

**Table 1 animals-10-01681-t001:** The proximate biochemical composition of diet used in the research.

Biochemical Composition	Content (%)
Moisture	8.8
Crude protein	46.2
Crude lipid	10.5
Ash	9.5

**Table 2 animals-10-01681-t002:** Primer sequences used in this study.

Primers *	Sequences (5′–3′)	Purpose
FADS2-F	CATCACGCCAAACCCAACATC	qRT-PCR
FADS2-R	AGACATAGACCAAGCCATATCCAC	qRT-PCR
Elovl5-F	GCTGAATATCTGGTGGTTCGTTATG	qRT-PCR
Elovl5-R	GGCTGGGATGGCTGAGAGG	qRT-PCR
β-actin-F	CAACTGGGATGACATGGAGAAG	qRT-PCR
β-actin-R	TTGGCTTTGGGGTTCAGG	qRT-PCR
SP1	AGCCGGATGCGGATGGAGAGAT	Promoter cloning
SP2	CCGTGGGCTGCCTACCTTGAGA	Promoter cloning
AP1	GTAATACGACTCACTATAGGGC	Promoter cloning
AP2	TCGACGGCCCGGGCTGGTAGCT	Promoter cloning
F-BF	GATTAGTATTTGGATATTGTAAGTTTTT	FADS2 BSP
F-BR	ACTTAAAAAAATTACACCATCTTACC	FADS2 BSP
E-BF	GGGAAAGTAGAGATAGTTTTGATGT	Elovl5 BSP
E-BR	ACTACCTACCTTAAAAAACRAAAAAAC	Elovl5 BSP

* Primers sequences for β-actin-F and β-actin-R were from [18]. Primers AP1 and AP2 were from the Universal GenomeWalker 2.0 kit (Clontech, 636405).

**Table 3 animals-10-01681-t003:** Muscle composition of Japanese sea bass cultured in marine cage and freshwater pond.

Composition	Freshwater	Seawater
Muscle lipid	2.0 ± 0.1 ^b^	2.8 ± 0.1 ^a^
Muscle protein	21.0 ± 0.1 ^b^	21.6 ± 0.2 ^a^
Muscle moisture	77.8 ± 0.2 ^a^	76.4 ± 0.3 ^b^

Within a row, different lower-case letters (a,b) indicate significant differences (*p* < 0.05) between the two groups.

**Table 4 animals-10-01681-t004:** Concentrations of various fatty acids in the muscle (mg/kg) ^1^. FP: freshwater pond; MC: marine cages.

Fatty Acid	FP Group	MC Group
C14:0	16.2 ± 7.7 ^b^	112.9 ± 20.6 ^a^
C16:0	714.1 ± 152.1 ^b^	1269.9 ± 322.9 ^a^
C18:0	247.1 ± 37.5 ^b^	341.4 ± 33.3 ^a^
∑SFA ^2^	977.4 ± 188.0 ^b^	1724.2 ± 149.4 ^a^
C16:1	102.4 ± 34.4 ^b^	296.3 ± 24.5 ^a^
C18:1n-9	98.7 ± 11.3 ^b^	187.9 ± 25.8 ^a^
∑MUFA ^3^	201.1 ± 93.6 ^b^	484.2 ± 97.1 ^a^
C18:2n-6	981.6 ± 179.2 ^b^	1312.5 ± 169.2 ^a^
C18:3n-6	30.1 ± 1.4 ^a^	25.8 ± 0.2 ^b^
C20:4n-6	92.4 ± 9.9 ^b^	167.1 ± 50.3 ^a^
∑n-6 PUFA ^4^	1104.2 ± 182.2 ^b^	1505.4 ± 151.7 ^a^
C18:3n-3	102.8 ± 12.7 ^b^	181.2 ± 27.4 ^a^
C20:5n-3	113.3 ± 14.7 ^b^	408.4 ± 70.0 ^a^
C22:6n-3	390.8 ± 54.4 ^b^	1294.9 ± 292.1 ^a^
∑n-3 PUFA ^5^	607.0 ± 67.5 ^b^	1884.5 ± 332.3 ^a^
∑n-3/∑n-6 PUFA	0.56 ± 0.08 ^b^	1.26 ± 0.25 ^a^
∑n-3 LC-PUFA ^6^	504.1 ± 68.3 ^b^	1703.3 ± 320.7 ^a^

^1^ Some fatty acids (e.g., C22:0, C24:0, C14:1, C20:2n-6 and C20:3n-6) were either not detected or present at very low concentrations; these fatty acids are not included in this table. ^2^ SFA: Saturated fatty acid. ^3^ MUFA: Monounsaturated fatty acid. ^4^ n-6 PUFA: n-6 polyunsaturated fatty acid. ^5^ n-3 PUFA: n-3 polyunsaturated fatty acid. ^6^ n-3 LC-PUFA: n-3 long-chain polyunsaturated fatty acid. Within a row, different lower-case letters (a,b) indicate significant differences (*p* < 0.05) between the two groups.

**Table 5 animals-10-01681-t005:** Concentrations of different fatty acids in the liver (mg/kg) ^1^.

Fatty Acid	FP Group	MC Group
C14:0	1205.9 ± 216.1 ^a^	902.7 ± 127.5 ^b^
C16:0	11,957.4 ± 1310.3 ^a^	9799.0 ± 1437.6 ^b^
C18:0	2987.9 ± 683.0 ^a^	1663.1 ± 104.5 ^b^
∑SFA ^2^	16,151.2 ± 992.2 ^a^	12,364.7 ± 340.9 ^b^
C16:1	4296.3 ± 586.1	3974.3 ± 540.4
C18:1n-9	1490.7 ± 111.3	1570.8 ± 272.6
∑MUFA ^3^	5787.0 ± 664.0	5555.0 ± 843.6
C18:2n-6	7066.9 ± 949.5 ^a^	2271.2 ± 254.3 ^b^
C18:3n-6	1211.9 ± 191.6 ^a^	896.7 ± 184.1 ^b^
C20:4n-6	605.9 ± 107.4	666.5 ± 110.5
∑n-6 PUFA ^4^	8884.7 ± 994.2 ^a^	3834.4 ± 263.0 ^b^
C18:3n-3	1217.2 ± 178.0 ^a^	741.0 ± 120.9 ^b^
C20:5n-3	410.5 ± 66.6 ^b^	986.3 ± 173.5 ^a^
C22:6n-3	1629.9 ± 226.1 ^b^	6386.0 ± 705.2 ^a^
∑n-3 PUFA ^5^	3257.5 ± 310.0 ^b^	8113.4 ± 740.8 ^a^
∑n-3/∑n-6 PUFA	0.4 ± 0.04 ^b^	2.1 ± 0.1 ^a^
∑n-3 LC-PUFA ^6^	2040.4 ± 253.1 ^b^	7372.3 ± 738.7 ^a^

^1^ Some fatty acids (e.g., C22:0, C24:0, C14:1, C20:2n-6 and C20:3n-6) were either not detected or were present at very low concentrations; these fatty acids are not included in this table. ^2^ SFA: Saturated fatty acid. ^3^ MUFA: Monounsaturated fatty acid. ^4^ n-6 PUFA: n-6 polyunsaturated fatty acid. ^5^ n-3 PUFA: n-3 polyunsaturated fatty acid. ^6^ n-3 LC-PUFA: n-3 long-chain polyunsaturated fatty acid. Within a row, different lower-case letters (a,b) indicate significant differences (*p* < 0.05) between the two groups.

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
