# Peer review of "Seawater Culture Increases Omega-3 Long-Chain Polyunsaturated Fatty Acids (N-3 LC-PUFA) Levels in Japanese Sea Bass (Lateolabrax japonicus), Probably by Upregulating Elovl5"

_animals, 2020, doi:10.3390/ani10091681_

Round 1

Reviewer 1 Report

The paper entitled Seawater culture increases omega-3 long-chain polyunsaturated fatty acids (n-3 LC-PUFA) levels in Japanese sea bass (Lateolabrax japonicus), probably by upregulating Elovl5 describes an approach to determine wether rearing fish of the same species either in seawater or in freshwater results in different levels of omega 3 LC PUFA.

Its a very interesting question which is very difficult to answer. My main concern with this study is about the fish used in this study. Some information is lacking in the Materials and Methods section:

  1. How were the fish reared? (Tanks? Sea cages? What size? How many fish per tank/cage?)

  2. Were the fish taken from the same broodstock? If not, the differences could be entirely due to some genetic differences between strains.

  3. At what temperature were the fish reared? How stable was the temperature?

  4. Light regime?

  5. Detailed diet table necessary. The one provided does not contain every important detail. (ingredients list)

Overall, the article needs a language check by a native speaker before it can be considered for publication.

Introduction

A bit short, could benefit from a better description of how this study adds information to the research field. Brief overview of other studies rather than multiple citation for one sentence.

Language check needed

Materials and Methods (see above for more points)

  1. More details on gas chromatography needed (could be supplementary material): Sample preparatioin? Freeze drying and then? Homegenization? Sonication? Centrifugation? How was the GC setup? Column length and thickness? Oven temperature regime? Sample volume? Detection method? Quantification? Internal standard?

  2. More details on qPCR. Even if the procedure was explained in another paper, provide an overview of how the qPCR was done here. Some information specific to this study is missing, i.e. How were the primers designed? Criteria? How was the reference gene chosen? Were more reference genes tested? Why only one reference gene? How stable was the reference gene? How were the samples analyzed on the plate? Were seawater and freshwater samples randomized on the same plate? Were there multiple plates per gene? If so, was a plate calibrator used to normalize between plates? How many technical replicates per sample? What was the variation between technical replicates?

  3. The cloning and sequencing is nicely described.

Since a lot of important information is lacking, I will not go into too much detail on the Results and Discussion sections.

Overall I think this article needs much more information to allow a proper review of the scientific soundness. The approach is interesting and I would gladly review it again, should it be re-submitted.

Author Response

Dear Sir,

Thank you very much for your very constructive comments, according to which our manuscript has been carefully revised.

Comment: The paper entitled Seawater culture increases omega-3 long-chain polyunsaturated fatty acids (n-3 LC-PUFA) levels in Japanese sea bass (Lateolabrax japonicus), probably by up regulating Elovl5 describes an approach to determine whether rearing fish of the same species either in seawater or in freshwater results in different levels of omega 3 LC PUFA.

It’s a very interesting question which is very difficult to answer. My main concern with this study is about the fish used in this study. Some information is lacking in the Materials and Methods section:

Response: Thank you very much for your positive comments on our manuscript. Japanese sea bass is extensively cultured in both freshwater and seawater in China, especially in the southern coastal region. It is widely known that culture water salinity may affect fish biochemical compositions, while many other environmental factors may also show the same effect in variety extend. Thus, it is hard to say whether water salinity played major roles in determining the different levels of omega-3 LC-PUFA content between the two groups. In the present study, we tended to treat freshwater pond culture and marine cage culture as two different culture patterns, trying not to attribute detected differences to a single environmental factor. As of the lack of some information, we have supplemented according to your comments.

Comment: How were the fish reared? (Tanks? Sea cages? What size? How many fish per tank/cage?)

Response: The freshwater reared fish was cultured in pond, which was about 1.5 acres in size, while the seawater group was raised in marine cages. Although the cages were much smaller in size (10 m*10 m*10 m) compared with freshwater pond, breeding density in the two culture patterns was comparable, approximately 6-8 per cubic metre. Detailed information has been added to the revised manuscript properly in line 106-114.

Comment: Were the fish taken from the same broodstock? If not, the differences could be entirely due to some genetic differences between strains.

Response: Yes, the two farms used fish fry from a common breeding company. This information has been added to our revised manuscript in line 108.

Comment: At what temperature were the fish reared? How stable was the temperature?

Response: To make it clear, we emphasize here that our experimental fish were sampled from freshwater and seawater commercial farms, where fish were cultured under nature conditions. The two farms were located in Shantou and Zhuhai, respectively, both located in the southern coastal region of China, where the annual average temperature ranges from 18 to 22 degree centigrade. When the fish was sampled in December 2018, local temperature was about 15 degree centigrade.

Comment: Light regime?

Response: The whole culture procedure was conducted in natural conditions, and no light regime was adopted.

Comment: Detailed diet table necessary. The one provided does not contain every important detail. (ingredients list)

Response: As we have mentioned in our manuscript, the diet that the two farms used were the same commercial diet, 8985 from TongWei company (Chengdu, China). Since we didn’t formulate the diet, detailed ingredients list was not necessarily supplied. We determined the proximate biochemical composition instead to show the general profile of the commercial diet used by the two farms (see figure 1).  

Comment: Overall, the article needs a language check by a native speaker before it can be considered for publication.

Response: Thank you for your kind suggestion. Actually, our manuscript has been checked and polished by LetPub, a sub branch of ACCDON Company in China, before we submitted it to Animals. To make sure of the writing fluency of our manuscript, we have invited another native speaker to check our revised manuscript before resubmitting.

Comment: The introduction is a bit short, and could benefit from a better description of how this study adds information to the research field. Brief overview of other studies rather than multiple citation for one sentence.

Response: We appreciative of your kind suggestion. The introduction part has been revised according to your advice. More information upon research in Japanese sea bass has been added in line 67-72. “Previous research has shown that seawater reared sea bass owned significantly higher eicosapentaenoic acids (EPA, 20:5n−3), docosahexaenoic acids (DHA, 22:6n−3) contents in both liver and muscle tissues [4]. To date, however, the underlying mechanism has not been well studied. Since the liver serves as the main organ responsible for fatty acids biosynthesis in fish, it is conjectured that biosynthesis of LC-PUFA is probably enhanced in seawater cultured sea bass compared with freshwater reared ones.” Both nutritional and environmental modulation of FADS2 and Elovl5 gene, and environmental factors induced epigenetic regulation on gene expression has been add in line 78-98. “The nutritional modulation of FADS2 and Elovl5 have been extensively researched to test whether desaturase and elongase activity could compensate for the reduction of tissue LC-PUFA levels due to supplementation of dietary vegetable oil [22, 23]. Some studies have focused on the regulatory mechanisms of FADS2 and Elovl5 genes. For instance, some transcription factors, Sterol Regulatory Element Binding Protein-1 (SREBP-1) and Peroxisome Proliferator-Activated Receptors (PPARs), regulate the two genes [24, 25]. In addition to nutritional factors, environmental factors have been demonstrated to mediate FADS2 gene expression regulation in fish. In Atlantic salmon, for instance, increase in LC-PUFA biosynthesis was observed during parr–smolt transformation via increasing FADS2 gene expression [26]. To date, however, it has been unclear whether Elovl5 is associated with environmental factors regulated LC-PUFA biosynthesis in fish, and, if so, how.”

“At present, many researches are focused on uncovering the mechanism underlying phenotypic variation in response to different environments. Growing evidence indicates that DNA methylation plays an important role in helping animals cope with different environments [27], which predominantly occurs at the 5-position of cytosine and plays a vital role in the epigenetic control of gene expression [28]. It was found that exposing adult zebrafish to estrogens could led to decrease in Vitellogenin gene promotor methylation level and hence increase gene expression [29]. In Half smooth tongue sole, water salinity stress may regulate gene expression via changing DNA methylation patterns at tissue-specific epigenetic loci [30]. Therefore, in this study, we aimed to answer the question that why seawater cultured sea bass owns higher EPA and DHA contents compared with freshwater cultured fish by focusing on key genes involved in LC-PUFA biosynthesis from the aspect of gene expression and epigenetic regulation.”

Comment: More details on gas chromatography needed (could be supplementary material): Sample preparation? Freeze drying and then? Homogenization? Sonication? Centrifugation? How was the GC setup? Column length and thickness? Oven temperature regime? Sample volume? Detection method? Quantification? Internal standard?

Response: We appreciative of your kind suggestion. A detailed work flow for sample preparation and GC analysis has been added in the METHODS section in line 126-133 and line 136-147, and we hereby provide this information to the reviewer. All samples subjected to fatty acids analysis were first freeze-dried at -40 °C for 48 hours, and then smashed and homogenized with a high-throughput and high-speed grinding machine. Moderate amount of sample was removed to hydrolysis in 10 mL hydrochloric acid solution with 100 mg gallic acid and 2 mL ethanol at 70-80 °C for 40 minutes. The hydrolysate was further extracted three times using ether and petroleum ether mixture and steamed to dry at 100±5 °C for 2 hours followed by saponification and esterification in alkaline solutions at 85 °C in water bath. When cool down to room temperature, 1 ml n-hexane was added to extract the esterification product, and 100μl supernatant was drawn and stabilized to 1 ml with n-hexane before filtration with 0.45μm membrane. The fatty acid composition was then determined with a gas chromatograph (7890A; Agilent), using a CNW CD-2560 chromatographic column (100 m × 0.25 mm × 0.20 μm). The heating procedure was as follows: 130 °C for 15 min, then raised to 240 °C at a rate of 4 °C /min for 30 min. The injector and detector temperatures were set to 250 °C and 260 °C, respectively, and the flow rate of carrier gas was 0.5 mL/min. The split ratio was 10:1. Fatty acid quantification by standard mixture of 35 fatty acid methyl esters (Sigma, CRM47885) and calculating formula:

W: every fatty acid content in sample (mg/kg).

C: the concentration of methyl esters of fatty acids (mg/L).

V: constant volume (mL).

N: dilution multiple.

k: conversion coefficient of each fatty acid methyl ester to fatty acid.

m: the weight of the sample (g).

Comment: More details on qPCR. Even if the procedure was explained in another paper, provide an overview of how the qPCR was done here. Some information specific to this study is missing, i.e. How were the primers designed? Criteria? How was the reference gene chosen? Were more reference genes tested? Why only one reference gene? How stable was the reference gene? How were the samples analyzed on the plate? Were seawater and freshwater samples randomized on the same plate? Were there multiple plates per gene? If so, was a plate calibrator used to normalize between plates? How many technical replicates per sample? What was the variation between technical replicates?

Response: We appreciative of your kind suggestion, and a general work flow for quantitative real time PCR has been added in the METHODS section in our revised manuscript in line 151-161. Briefly, the concentration of the prepared total RNA was determined and 1 μg of total RNA was treated with gDNA Eraser (Takara, Dalian, China) to remove possible DNA contaminants according to the manufacturer’s instructions followed by reverse transcription using PrimeScript RT reagent kit (Takara). Quantitative real-time PCR was performed on AB 7500 Fast platform (Applied Biosystems, Carlsbad, California) according to the instructions provided in the fluorescence quantitative PCR kit (Takara). Primers for β-actin, FADS2, and Elovl5 were designed using Oligo software v.7 based on mRNA sequences downloaded from GenBank database (Table 2). Parameters adopted including oligo length 18-26 nt, GC content 30%-70%, and avoided high terminal stability, hairpin structure, dimer complex and false priming. A couple of commonly used genes, e.g., β-actin, 18s rRNA, GAPDH, and ef1α have been tested for expression stability in our former study (cited in our manuscript as reference 18), where β-actin showed most stable expression in different tissues. Therefore, β-actin was chosen as an internal standard to normalize expression level of FADS2 and Elovl5. Amplification efficiencies for β-actin, FADS2, and Elovl5 were 95.7%, 101.2% and 98.3%, respectively, determined by quantitative real time PCR analysis on gradient diluted cDNA. Five biological replicates and two technical replicates were conducted to minimize random error. However, we did not randomize samples on the plate because it would perplex the handling process and easily led to mistakes. Since the samples were limited, we put both biological and technical replicate of each gene on the same plate to avoid the possible inter-plate error. Moreover, all reagents used in quantitative real-time PCR except cDNA were premixed before use to minimize sampling error. The difference of Ct values between our technical replicates were all less than 1, and mostly less than 0.5. Given some of the mentioned principles used in primer designing and reaction preparation are ordinary and compulsory in conducting quantitative real-time PCR, these information are not added in our revised manuscript.

Best regards,

Xuedi Du

Reviewer 2 Report

The manuscript entitled “Seawater culture increases omega-3 long-chain polyunsaturated fatty acids (n-3 LC-PUFA) levels in Japanese sea bass (Lateolabrax japonicus), probably by upregulating Elovl5”, written by Xiaojing Dong, Jianqiao Wang, Peng Ji, Longsheng Sun, Shuyan Miao, Yanju Lei and Xuedi Du is in general a well written paper which provides interesting information on the effects of two different rearing salinities (0.1 and 26.2 g/L) on the long chain polyunsaturated fatty acid compositions of the muscle and liver in Lateolabrax japonicus raised either in freshwater or in seawater.

In Japanese sea bass, as it is described in the present study, the seawater culture increase the content of n-3 LC-PUFA. Furthermore, quantitative real-time PCR analysis showed that seawater rearing led to enhanced fatty acid elongase 5 (Elovl5) transcript abundance but decreased fatty acid desaturase 2 (FADS2) expression levels compared with freshwater culture. Moreover, two of the CpG loci in the FADS2 gene promoter region were heavily methylated in the seawater-cultured fish, while the methylation patterns of Elovl5 gene promoter region showed no significant difference between the two groups.

Thus, the authors suggest that Elovl5, but not FADS2, plays an important role in the regulation of LC-PUFA synthesis under seawater conditions. However, the mechanisms underlying this effect is not described, which is my major concern.

Several concerns

Keywords

According to my opinion the keywords should not overlap with the title, although several journals have no such specifications.

Material and Methods

The authors do not provide detailed information on the raised conditions in the two different farms, besides that the fish were fed the same commercial diet and their body size was almost similar (between 400-500 g). One should have expected more information both on different raised abiotic factors such as the water temperature, or the oxygen content, or even the different genetic background or even the embryogenesis of the fish raised in the different farms, which might affect their fatty acid biosynthesis.

Line  113: L. japonicas in italics

Discussion

Lines 223-224 and 234: S. canaliculatus in italics

Line 246: “the DNA methylation states of the CpG dinucleotides in the gene promoter regions of both genes were determined using BSP. We found that two CpG loci in the FADS2 gene promoter region were heavily methylated in the seawater group, but only slightly methylated in the freshwater group. “ This is a novel and interesting finding. Do the authors study whether these changes (DNA methylation) were associated with differences in the mRNA expression of the examined genes in liver and/or muscle from farmed fish by calculation of Pearson's correlation coefficient? 

References

Lines 328-329: Tocher, D.R., Omega-3 long-chain polyunsaturated fatty acids and aquaculture in perspective. Aquaculture, 2015.

Please, add Aquaculture 449, pp. 94-107 (https://doi.org/10.1016/j.aquaculture.2015.01.010)

Lines 364-365:

Al-Lawati, A., S. Al-Bahry, R. Victor, et al., Salt stress alters DNA methylation levels in alfalfa (Medicago  spp). Genetics and Molecular Research, 2016. 15(1)

Please, add Genetics and Molecular Research, 2016. 15(1), p. 15018299.

 (doi: 10.4238/gmr.15018299)

Author Response

Dear Sir,

We are appreciative of your constructive comments, according to which we have carefully revised our manuscript.

Comment: The manuscript entitled “Seawater culture increases omega-3 long-chain polyunsaturated fatty acids (n-3 LC-PUFA) levels in Japanese sea bass (Lateolabrax japonicus), probably by upregulating Elovl5”, written by Xiaojing Dong, Jianqiao Wang, Peng Ji, Longsheng Sun, Shuyan Miao, Yanju Lei and Xuedi Du is in general a well written paper which provides interesting information on the effects of two different rearing salinities (0.1 and 26.2 g/L) on the long chain polyunsaturated fatty acid compositions of the muscle and liver in Lateolabrax japonicus raised either in freshwater or in seawater.

In Japanese sea bass, as it is described in the present study, the seawater culture increases the content of n-3 LC-PUFA. Furthermore, quantitative real-time PCR analysis showed that seawater rearing led to enhanced fatty acid elongase 5 (Elovl5) transcript abundance but decreased fatty acid desaturase 2 (FADS2) expression levels compared with freshwater culture. Moreover, two of the CpG loci in the FADS2 gene promoter region were heavily methylated in the seawater-cultured fish, while the methylation patterns of Elovl5 gene promoter region showed no significant difference between the two groups.

Thus, the authors suggest that Elovl5, but not FADS2, plays an important role in the regulation of LC-PUFA synthesis under seawater conditions. However, the mechanisms underlying this effect is not described, which is my major concern.

Response: Thank you very much for your positive comments on our manuscript. Since our experimental fish were sampled from commercial farms culturing sea bass in freshwater and seawater respectively, it is hard to attributes the detected differences between the two groups to water salinity difference, although which was the main different environmental factor between the two groups. We prefer to define our study a comparative research on different culture patterns, i.e., freshwater pond culture and marine cage culture.

As of the mechanism underling the effect of Elovl5 on LC-PUFA synthesis, we believe it is quite complicated. Not only gene promoter activity, regarding the methylation pattern of the CpG island, but also many different transcription factors, for instance PPAR, might contributes to gene expression regulation of Elovl5, directly or indirectly. Thus, more research is needed to reveal the very mechanism underlying this effect. Actually, we are designing new experiments to answer this question by conducting a well-controlled feeding trial under different water salinity followed by transcriptome sequencing and molecular verification. Thus, in the present study, we did not intend to give the mechanism underlying this effect.

Comment: According to my opinion the keywords should not overlap with the title, although several journals have no such specifications.

Response: Thank you for your kind suggestion. The keywords has changed to “sea bass; freshwater pond culture; marine cage culture; n-3 LC-PUFA; gene promoter methylation” in line 49-50.

Comment: The authors do not provide detailed information on the raised conditions in the two different farms, besides that the fish were fed the same commercial diet and their body size was almost similar (between 400-500 g). One should have expected more information both on different raised abiotic factors such as the water temperature, or the oxygen content, or even the different genetic background or even the embryogenesis of the fish raised in the different farms, which might affect their fatty acid biosynthesis.

Response: We are sorry for the absence of several important information regarding the breeding conditions and experimental fish adopted by the two farms. To make it clear, we have supplemented necessary information in our revised manuscript in line 106-114. As our experimental fish were sampled from commercial farms in the southern coastal region of China, where the annual average temperature ranges from 18 to 22 degree centigrade, after eight months of culture under nature conditions, both water temperature and dissolved oxygen flux in certain ranges in both freshwater pond and seawater cages. It is certain that difference in temperature and dissolved oxygen exists during the whole feeding period between the two culture patterns because freshwater pond culture is totally different from marine cage culture, where many environmental factors such as dissolved oxygen could hardly be artificially regulated. Since the experimental fish was not cultured in fine controlled experimental tanks, we tended to attribute the difference in body composition to the different culture patterns but not specific environmental factors. According to the farm managers, both farms used the same fish fry from a common breeding company. Therefore, we can infer from our study that the different culture pattern, i.e., freshwater pond culture and seawater cage culture, is the major factor led to the biochemical difference between the two groups.

Best ragards,

Xuedi Du

Round 2

Reviewer 1 Report

Dear Authors,

I would like to commend you for your thorough review. The manuscript certainly improved a lot and I would now recommend it for publication. It reads much better and the necessary information was kindly provided.

Kind regards

This manuscript is a resubmission of an earlier submission. The following is a list of the peer review reports and author responses from that submission.